# Making More Womb: Clinical Perspectives Supporting the Development and Utilization of Mesenchymal Stem Cell Therapy for Endometrial Regeneration and Infertility

**DOI:** 10.3390/jpm11121364

**Published:** 2021-12-14

**Authors:** Michael Strug, Lusine Aghajanova

**Affiliations:** Division of Reproductive Endocrinology and Infertility, Department of Obstetrics and Gynecology, School of Medicine, Stanford University, Sunnyvale, CA 94087, USA; mstrug@stanford.edu

**Keywords:** endometrium, mesenchymal stem cells, bone marrow mesenchymal stem cells, infertility, Asherman’s syndrome

## Abstract

The uterus is a homeostatic organ, unwavering in the setting of monthly endometrial turnover, placental invasion, and parturition. In response to ovarian steroid hormones, the endometrium autologously prepares for embryo implantation and in its absence will shed and regenerate. Dysfunctional endometrial repair and regeneration may present clinically with infertility and abnormal menses. Asherman’s syndrome is characterized by intrauterine adhesions and atrophic endometrium, which often impacts fertility. Clinical management of infertility associated with abnormal endometrium represents a significant challenge. Endometrial mesenchymal stem cells (MSC) occupy a perivascular niche and contain regenerative and immunomodulatory properties. Given these characteristics, mesenchymal stem cells of endometrial and non-endometrial origin (bone marrow, adipose, placental) have been investigated for therapeutic purposes. Local administration of human MSC in animal models of endometrial injury reduces collagen deposition, improves angiogenesis, decreases inflammation, and improves fertility. Small clinical studies of autologous MSC administration in infertile women with Asherman’s Syndrome suggested their potential to restore endometrial function as evidenced by increased endometrial thickness, decreased adhesions, and fertility. The objective of this review is to highlight translational and clinical studies investigating the use of MSC for endometrial dysfunction and infertility and to summarize the current state of the art in this promising area.

## 1. Introduction

The endometrium is both unique and fascinating in its ability to undergo regular cycles of growth, followed by vasoconstriction, hypoxia, cell death, tissue desquamation, shedding, followed by scar-less wound healing and regeneration, with angiogenesis serving an essential role [1,2,3,4]. Endometrial tissue is quite forgiving with the capacity to regenerate even after massive shedding of its functional layer, such as following parturition. The key to successful repair, as in any other tissue, is the preservation of not only resident endometrial epithelial and stromal cells in the basalis layer, but also the presence and survival of the endometrial stem cell population [5].

While the process of endometrial repair is overall efficient, significant insults to the regenerating layer of endometrium can result in endometrial pathologies such as intrauterine adhesions or endometrial atrophy, clinically presenting as oligomenorrhea/amenorrhea, infertility, and pregnancy loss [6]. These patients are typically referred to fertility clinics, hence the issues are most frequently encountered by infertility specialists. While therapeutics such as *in vitro* fertilization (IVF), intracytoplasmic sperm injection, and pre-implantation genetic testing bypass infertility resulting from male factor, tubal, ovarian, or even genetic factors, our understanding of endometrial contributors to infertility and therefore their treatment success remains limited. Endometrial plasticity and self-regeneration are essential for supporting embryo implantation, and reproductive consequences, such as infertility and recurrent pregnancy loss, are noted in their absence [7,8]. Therefore, novel therapeutics for treating endometrial-driven infertility are necessary, and stem cells, including endometrial, represent a promising area for innovation in Reproductive Medicine.

Mesenchymal stem cells (MSC) were first described in bone marrow isolates based on their self-regenerating capacity, ability to grow in culture on plastic, and capacity to differentiate towards other cell types within the mesodermal lineage [9]. These characteristics, along with the presence of MSC markers (CD105, CD73, CD90) and absence of hematopoietic markers (i.e., CD45, CD34, HLA-DR), are maintained as criteria for defining MSC [10]. Other tissue sources for MSC have been described extensively, including adipose tissue [11], the umbilical cord [12], and the placenta [13]. Since the beginning of the 21st century, studies have sought to characterize endometrial-derived stem cells and their role in the human endometrium [5,14]. Based on these initial studies, significant efforts have been made towards understanding endometrial progenitors, side populations, and stem cells of endometrial, decidual, and menstrual origin [5,15]. Bone marrow-derived mesenchymal stem cells (bmMSC) and endometrial and menstrual mesenchymal stem cells (eMSC and MenSC, respectively) are among the most well described. MSC react to local factors secreted in response to tissue injury to promote tissue regeneration through promoting proliferation while preventing apoptosis and fibrosis [16]. Furthermore, their immunomodulatory functions and low immunogenicity make them a suitable candidate for therapeutic purposes. While still in the early stages of development, several small studies have highlighted the potential implications for the use of both endometrial derived and non-endometrial MSC for clinical conditions associated with abnormal endometrial function. The purpose of this review is to recap the need for endometrial stem cell therapy and summarize the data supporting its potential clinical use, primarily in the setting of conditions impacting infertility.

## 2. Clinical Conditions Associated with Endometrial Dysfunction and Infertility

### 2.1. Asherman’s Syndrome

Asherman’s Syndrome (AS) represents the most notable and well-characterized clinical condition associated with endometrial dysfunction and resulting intrauterine adhesions. AS is defined by clinical symptoms such as menstrual irregularities, amenorrhea, dysmenorrhea, pelvic pain, and infertility, in the setting of intrauterine adhesions [17,18]. While the clinical sequelae of intrauterine adhesions may vary and are not typically too serious, the physical, psychological, and emotional burden on affected women is significant [19]. Pregnancy and uterine instrumentation are among the most common risk factors associated with the development of Asherman’s Syndrome. In a meta-analysis of 10 prospective studies including 1770 women with a hysteroscopic evaluation within 12 months of a miscarriage, approximately 19% developed intrauterine adhesions [20]. The risk for adhesion formation increased with the number of miscarriages and whether instrumentation with dilation and curettage (D & C) was performed.

Interestingly, subsequent studies have indicated larger uterine size and method for D & C (mechanical suction rather than manual vacuum aspirator) influence the development of intrauterine adhesions [21]. In a cohort of AS patients undergoing operative hysteroscopy, a history of first-trimester instrumentation (58%) was more likely than postpartum instrumentation (38%); however, the extent of adhesive disease was more severe after postpartum instrumentation [22]. Common gynecologic procedures such as hysteroscopic uterine septum resection and myomectomy (abdominal or hysteroscopic) also may predispose to AS [23]. Non-surgically induced intrauterine adhesions may develop following infectious/inflammatory conditions, such as *Mycobacterium tuberculosis* or chronic endometritis [24,25]. The history of a levonorgestrel-intrauterine device (IUD) may present a rare, but potentially uprising cause of intrauterine synechia [26]. Remarkably, synechiae may develop without an identifiable risk factor and are discovered during routine hysteroscopic evaluation [27].

In cases of AS, the common clinical management is directed towards hysteroscopic removal of intrauterine adhesions, prevention of their recurrence, and reformation [28,29,30]. While adhesiolysis may appear as a sound management strategy, the treatment often fails particularly in subjects with severe intrauterine adhesions and significant clinical presentation such as amenorrhea [29]. In a 10-year retrospective cohort study, up to three surgical procedures were required to restore menses and endometrial cavity anatomy in 95% of patients and there was a 29% risk of adhesion recurrence noted [22]. Surgical management does enhance fertility, where approximately half of women will conceive following adhesiolysis, but, unfortunately, pregnancy rates remain relatively low in women with severe disease (25%) [31]. Additionally, endometrial cells in AS lose their responsiveness to estrogen and progesterone, suggesting a functional deficit in addition to the physical impact of adhesions [5].

### 2.2. Persistent Thin Endometrium and Endometrial Atrophy

Unexplained persistent atrophic/thin endometrial lining represents another major clinical challenge in women with infertility and may be associated with recurrent implantation failure in IVF cycles [32]. An endometrial thickness less than 7 mm in an IVF cycle confers significantly lower clinical pregnancy and live birth rates and increases the risk for miscarriage [33,34]. Thin endometrium may be present in the setting of AS, in response to long-term hormonal contraception use, or may be idiopathic [26,35]. Standard first-line management of thin endometrium during assisted reproduction cycles is estradiol supplementation via either oral, transdermal, intramuscular, or vaginal preparations [36]. Some studies have shown extended estradiol supplementation alone may improve endometrial thickness and pregnancy rates [37], while others have displayed conflicting results [38]. While there are proposed adjuvant treatment options for thin endometrium such as vitamin E, pentoxifylline, L-arginine, aspirin, tamoxifen, and sildenafil, many of these have not been definitively shown to improve endometrial thickness and fertility outcomes [39]. Autologous platelet-rich plasma treatment represents another promising therapeutic for thin endometrium and recurrent implantation failure, but larger studies are needed to validate its efficacy [40]. These dire situations (AS and thin endometrial lining associated with infertility and implantation failure) present a tremendous psychological trauma to patients and force them towards extreme options such as considering the use of gestational carrier (surrogacy), uterine transplant, or adoption, with its significant financial and psychological burden and often being merely unaffordable.

## 3. Clinical Foundation for Stem Cell Use for the Treatment of Endometrial-Related Infertility

As mentioned earlier, it is well recognized that the presence of resident or stem cells in the basalis layer of the endometrium is a prerequisite for successful tissue regeneration. The endometrium will regenerate in response to endogenous or exogenous estrogen, and alternative experimental treatments for promotion of endometrial regeneration have been proposed with limited positive evidence, such as infusion of platelet-rich plasma, human chorionic gonadotropin, or granulocyte colony-stimulating factor [30,41,42]. However, when the regenerative layer is severely damaged and replaced with scar tissue with little or no healthy cells left, relying on the regenerative capacity of the endometrium may be pointless. Thus, exogenous autologous or allogeneic stem cells may prove to be the solution.

### 3.1. Bone Marrow-Derived Mesenchymal Stem Cells

Bone marrow-derived mesenchymal stem cells (bmMSC) originate from bone marrow aspirates and represent a small fraction of the total population of stromal cells collected, ranging from 0.002 to 0.02%, and the yield decreases with advancing age [43,44]. Additionally, bmMSC are associated with the shortest culture period and lowest proliferation capacity compared to other MSC sources [45]. Cell surface markers associated with bmMSC include CD44, CD73, CD90, CD105, and CD166 with absence of hematopoietic markers, CD45, CD34, and CD14 [46]. Bone marrow-derived mesenchymal stem cells have been long entertained as a possible source of endometrial regeneration [47,48]. Following HLA-mismatched bone marrow transplants, recipients were noted to have donor bone marrow-derived epithelial and stromal cells [47]. Subsequent studies in female recipients of male bone marrow confirmed bone marrow contribution to endometrial epithelial and stromal cells as noted by the presence of Y chromosome in endometrial cells [49]. The mechanisms for bmMSC recruitment to the endometrium were largely determined through mouse studies rather than human data. In a mouse bone marrow transplantation model, bmMSC were recruited to the endometrial stroma in response to uterine ischemia/reperfusion, and this recruitment was independent of hormonal cycling [50]. A mouse mechanical endometrial injury “AS model” displayed increased recruitment of donor bmMSC to the epithelium and stroma [51]. They also noted improved pregnancy rates in mice who had received intravenous bone marrow transplantation.

Nagori et al. 2011 described the first report of the clinical use of autologous bone marrow-derived MSC for treatment of AS [52]. In a patient with severe AS and infertility, they performed intra-uterine instillation of autologous BM-derived CD9, CD44, and CD90-positive cells after hysteroscopic treatment of synechiae followed by cyclic estradiol and medroxyprogesterone. Following treatment, there were improvements in endometrial thickness, morphology, and vascularity noted on ultrasonography, and the patient ultimately conceived following donor oocyte-derived embryo transfer [52]. Singh et al. performed sub-endometrial injection of autologous bone-marrow-derived mononuclear cells in a case report of six amenorrheic women with AS refractory to standard management with hysteroscopic adhesiolysis (two to three times) and oral estrogen therapy [53]. Five out of six women had a history of genital tuberculosis and one had a history of D & C. Menses returned in five out of six patients when followed up to nine months. Endometrial thickness in these patients was improved; however, no report on subsequent pregnancy outcomes was provided. More recently, Singh et al. published a follow-up study to their initial report including 25 patients with refractory AS (n = 12) or endometrial atrophy (n = 13), and up to 15 patients were followed up to five years following sub-endometrial injection of autologous bone-marrow-derived mononuclear cells [54]. Similar to their first study, a large proportion of patients in each group had a history of genital tuberculosis (six in each group). Menses were restored in amenorrheic patients (six out of seven), and improvements in endometrial thickness occurred but plateaued after three months post-procedure. Out of 15 patients followed through five years, three patients conceived (two spontaneous, one via IVF) with live births, and one patient developed an ectopic pregnancy [54].

In a small cohort study of 16 women (11 with AS, 5 with endometrial atrophy), autologous CD133+ bmMSC were injected into the uterine artery and spiral arterioles via intra-arterial catheterization, followed by hormonal therapy [55]. Follow-up ultrasound, hysteroscopy, and endometrial biopsies were performed. Menses resumed in all but one patient in the first month following treatment, but the length of menses waned with subsequent cycles. There were improvements in AS disease scoring based on second-look hysteroscopy, and endometrial thickness was improved in the setting of both AS and endometrial atrophy. They also measured increased neo-angiogenesis defined by the presence of mature blood vessels (positive for CD31 and α-SMA by immunofluorescence). Pregnancy outcomes were reported with three spontaneous conceptions resulting in a live birth, ongoing pregnancy, and a miscarriage; seven pregnancies following 14 embryo transfers with only one live birth and one ongoing pregnancy, the remaining resulting in pregnancy losses and ectopic pregnancy [55]. These studies, while intriguing and innovative, provide a limited critical evaluation of the role for bm-MSC as they are case reports or case series lacking statistical power or control patients to determine true effects. Further, the specific markers for stem cell isolation have been variable.

### 3.2. Menstrual Mesenchymal Stem Cells

Menstrual blood is considered another source of mesenchymal stem cells. Menstrual fluid collected from menstrual cups and cultured on plastic yields menstrual mesenchymal stem cells (MenSC). MenSC shares some similarities with both bmMSC and eMSC and can serve as another source of mesenchymal stem cells with differentiation potential across mesodermal, ectodermal, and endodermal lineages [56]. Interestingly, MenSC is associated with higher yield (around 2–4-fold higher) and proliferative capacity compared to bmMSC [57,58]. Cell-surface markers for MenSC include CD29, CD44, CD73, CD90, CD105, and CD166 with an absence of hematopoietic markers, similar to other MSC [5]. They also are positive for eMSC markers CD146 and CD140b, reviewed further in the subsequent section. Like other MSC, MenSC contains regenerative and immunomodulatory properties with potential implications for restoring endometrial function in the setting of pathologic states. Additionally, they can be decidualized in culture, as confirmed by the expression of decidual markers prolactin and insulin-like growth factor binding protein-1 [59]. Furthermore, the secretion of decidual factors is increased in MenSC as compared to decidualized bmMSC and amniotic fluid-derived stem cells [60]. These inherent characteristics of MenSC suggest multiple clinical applications, particularly in the setting of endometrial dysfunction and infertility.

Zhang et al. performed one of the first studies to investigate a role for human MenSC during endometrial repair in a mouse model of electrocoagulation injury [61]. They found improvement in endometrial thickness, microvessel density, and restored fertility with MenSC administration. Paired in vitro studies using cultured media from MenSC showed reduced hydrogen peroxide-induced injury in human umbilical vein endothelial cells [61]. Other animal studies have since described MenSC-driven endometrial repair in a mouse endometrial mechanical injury model [62]. In another study, Zheng et al. highlighted the capacity of human MenSC-derived endometrial cells to form endometrial tissue using an *in vivo* transplantation model of MenSC placed in the axillary subcutaneous tissue of immunocompromised mice [63].

The clinical utility of MenSC for the treatment of severe AS has been evaluated in a few clinical studies. Tan et al. performed one of the earliest known studies in autologous MenSC transplantation [64]. In their non-controlled, prospective study, seven patients with severe AS and infertility received autologous intrauterine MenSC transplantation following an endometrial scratch on day 16 of the same cycle with repeat therapy if favorable endometrial growth did not occur. Menstrual blood was collected via a catheter on day two of menses followed by washing in antibiotic and heparin containing phosphate-buffered saline. Isolated mononuclear cells were then cultured for 14 days and the supernatant was tested to ensure the absence of infectious agents prior to intrauterine infusion of 0.5 mL containing 1 × 10^6^ cells. Embryo transfers were performed once desirable endometrial thickness was achieved (>7 mm). One patient spontaneously conceived and two out of four patients who underwent embryo transfer successfully conceived, but their pregnancy outcomes were not clearly reported [64].

More recently, Ma et al. reported improved endometrial thickness in 12 infertile women with refractory AS along with the increased duration of menses following autologous MenSC transplantation [65]. The aforementioned study by Zheng et al. found reduced cloning efficiency and OCT-4 expression from MenSC derived from women with severe AS [63], suggesting a possible existing abnormality within these progenitor cells and may begin to explain why only some but not all women develop severe adhesions under similar circumstances. This is in contradiction with the reports on improvement in endometrial regeneration in women with AS after autologous MenSC transplantation [64], but it is possible that the capacity to increase cell numbers for transplantation may overcome these barriers. Of note, some studies have highlighted the presence of OCT-4 expression within MenSC [66,67], while MSC obtained from other sources (i.e., BM, adipose, placental) do not express OCT-4 [68]. However, OCT-4 expression was cytoplasmic in MenSC, suggesting that regenerative potential for MSC (including MenSC) are less likely related to pluripotency but rather as a result of local tissue anti-fibrotic and immunomodulatory effects [5,67]. Interestingly, platelet-rich plasma supplementation in combination with MenSC synergistically improved endometrial repair and fertility outcomes in a rat endometrial mechanical injury model [69]. Altogether, MenSC represent a promising and readily available, potentially autologous non-invasive source for treating endometrial causes of infertility; however, more clinical trials on safety and efficacy are necessary to understand their potential. Further, streamlining the process for easier application in the clinical setting will need to be addressed as well.

### 3.3. Endometrial Mesenchymal Stem Cells

Endometrial mesenchymal stem cells (eMSC) are located peri-vascularly in both the endometrial functional and basal layers, and markers include CD140b and CD146 [70] or single marker SUSD2 [71]. Based on their localization, eMSC can be isolated from either an endometrial biopsy or menstrual blood and can re-capitulate other multi-lineage tissues *in vitro*. Endometrial biopsies contain eMSC, although with high variability [72,73]. CD140b/CD146 dual-positive cells represent approximately 0.04–19% and 0.8–15.7% of sorted stromal cells from hysterectomy and endometrial biopsy specimens, respectively, and FACS sorting resulted in cells with colony-forming efficiencies 0.2 to 4% [73]. We are not aware of studies on the clinical use of eMSC for reproduction or infertility, as most have evaluated MenSC, which are more readily accessible in larger quantities. There is, however, a large amount of work is building possibilities and expectations for clinical utilization of eMSC. Some studies have evaluated the potential for eMSC to regenerate endometrial tissue using an extracellular matrix scaffold of decellularized uteri and endometrial tissue from multiple species, ranging from rodents to women [74,75]. Primary endometrial cells used for recellularization of decellularized human endometrial tissue were hormonally sensitive and able to decidualize in vitro [75]. Moreover, stem cell-derived tissues reportedly engrafted to a partially excised rat uterus with subsequent pregnancies, although placental adherence was not noted within the graft [74]. Kuramoto et al. assessed the application of rat green fluorescent protein (GFP)-labeled primary endometrial stromal and epithelial cell sheets following surgical excision of the endometrium [76]. They found GFP-labeled cells integrated throughout the re-formed endometrium, and pregnancies did occur in the transplanted areas.

Clinical applications for eMSC in infertility are mostly theoretical at this point given the lack of human studies. There is an increasing trend to study endometrial tissue cell-to-cell interactions *in vitro* through the development of organoids in the setting of physiologic and pathologic states [77]. The application of endometrial tissue scaffolds or organoids may serve to regenerate the endometrium in women with conditions such as AS and atrophic endometrium, thus enhancing fertility. In addition, honing the immunomodulatory functions of eMSC in tissue scaffolds or organoids for use at the time of surgical interventions associated with adhesion formation, such as myomectomy, polypectomy, or uterine septum resection may aid in increasing regeneration and preventing adhesion development. More translational studies assessing the safety and utility of eMSC for clinical use are necessary.

### 3.4. Alternative Sources for Mesenchymal Stem Cells for Treatment of Infertility

#### 3.4.1. Umbilical Cord Mesenchymal Stem Cells

Other possible sources of putative multipotent stem cells have been explored for endometrial regeneration in the setting of endometrial dysfunction. Umbilical cord-derived mesenchymal stem cells (UC-MSC) are multipotent mesenchymal stem cells, which display similarities to bmMSC and are easily obtained from umbilical cord remnants that would otherwise be discarded following birth [78]. UC-MSC can be isolated from various components of the umbilical cord including the umbilical cord blood, umbilical cord lining, subendothelial layer, perivascular zone, or Wharton’s Jelly [79]. Markers for UC-MSC are similar to other MSC but they are negative for CD133 [79]. UC-MSC contain significantly higher proliferative capacity than bmMSC and differentiate towards endometrial epithelial and stromal cells, which can be decidualized with secretion of prolactin and insulin-like growth factor binding protein-1 expression [80]. Zhang et al. studied the role of UC-MSC in a rat model of endometrial injury [81]. The ethanol-induced endometrial injury was ameliorated following tail-vein injection of UC-MSC. Endometrial cell-specific markers (vimentin and cytokeratin) along with markers of proliferation and angiogenesis were upregulated, while inflammatory markers interferon-γ and tumor necrosis factor (TNF)-α were downregulated with treatment. Additionally, UC-MSC administration partially restored fertility in those rats [81]. Xin et al. performed *in vitro* and *in vivo* studies of collagen scaffolds containing human UC-MSC and its impact on human endometrial stromal cells and rat endometrial repair [82]. UC-MSC collagen scaffolds increased vascular endothelial growth factor-A, transforming growth factor-β1 and platelet-derived growth factor protein production in both a co-culture model with human endometrial stromal cells and in their rat endometrial injury model. In the rat model, UC-MSC collagen scaffolds had fused by five days post-transplantation and resulted in improved endometrial thickness and partially restored pregnancy rates [82].

There have also been small clinical studies describing the role of UC-MSC in women with thin endometrium secondary to AS [83,84]. In a study of 26 infertile women with refractory AS, the placement of a collagen scaffold loaded with UC-MSC following an adhesiolysis procedure restored endometrial thickness and reduced adhesion reformation on second-look hysteroscopy [83]. Histologic analysis revealed increased levels of estrogen receptor α, proliferative (Ki-67), vascular (von Willebrand Factor), and stromal cell (vimentin) markers. Intriguingly, short tandem repeat analysis only demonstrated patient DNA in sampling of the regenerated endometrium. At the end of the 30-month follow-up period, eight out of the 26 patients had live births. There were no serious adverse reactions noted [83]. In a more recent study, 16 infertile women with canceled embryo transfer cycles due to a thin endometrium (≤5.5 mm) received two transplants of UC-MSC collagen scaffolds [84]. They found a significant increase in average endometrial thickness from 4.08 to 5.87 mm. Following treatment, two patients had live births after a frozen embryo transfer, one patient had preterm birth at 25 weeks gestation after a frozen embryo transfer, and one patient had a live birth from an unassisted conception.

#### 3.4.2. Amniotic-Derived Mesenchymal Stem Cells

Placental-derived human amniotic cells (epithelial and mesenchymal) exhibit stem cell properties with low immunogenicity or tumorigenesis potential [85,86]. MSC isolated from placental/fetal membranes displays similar surface marker expression (positive for CD73, CD90, and CD105, while negative for hematopoietic markers CD34 and CD45) and differentiation potential to other MSC [87]. Yield from term amnion may be as high as 5 × 10^8^ MSC [88]. Similar to other adult stem cells previously mentioned, studies were developed using rodent intrauterine adhesion models to explore the therapeutic potential for amniotic epithelial and mesenchymal cells [89,90,91,92]. These animal studies reported both cell types improved endometrial thickness and glandular development with decreased fibrosis. Amniotic cells decreased the expression of pro-inflammatory cytokines, such as TNF-α, following endometrial injury [89]. In one of the mouse intrauterine adhesion models, fertility was improved following the transplantation of human amniotic epithelial cells [90]. In the same publication, *in vitro* studies were performed to assess the impact of co-culture of human amniotic epithelial cells with human endometrial stromal cells during hydrogen peroxide-induced injury. The presence of human amniotic epithelial cells resulted in enhanced autophagy, a developmental process important for reproductive tract homeostasis [93]. Given the ease and availability of placental tissue and fetal membranes, which are typically discarded following delivery, it is surprising that there have been no published clinical studies evaluating a role for amniotic-derived stem cells in the human endometrium.

#### 3.4.3. Adipose-Derived Mesenchymal Stem Cells

Adipose-derived mesenchymal stem cells (adMSC) display similar multi-lineage differentiation potential as bone marrow-derived MSC and are typically obtained through liposuction [94]. Interestingly, from the same amount of tissue, approximately 500 times more adMSC are obtained compared to bmMSC [95]. Additionally, adMSC display greater proliferative capacity than bmMSC [45]. Specific markers distinguishing adMSC from bmMSC include CD10, CD36, and CD106 [96]. The use of adMSC has been evaluated in several mouse models of AS [97,98,99]. Kilic et al. published the first study of adMSC in a rat AS model where the endometrial injury was induced with trichloroacetic acid [97]. Local injection of adMSC into the damaged uterine horn was performed with subsequent intra-peritoneal adMSC injections with or without exogenous oral estrogen supplementation. adMSC injection alone increased cellular and vascular proliferation but did not reduce fibrosis. Administration of estrogen alone or in combination with adMSC synergistically reduced fibrosis. However, the impact on fertility in response to treatment was not reported [97]. Shao et al. labeled adMSC with GFP prior to local uterine administration to a rat ethanol-induced endometrial injury model [98]. They reported improved endometrial thickness, microvessel density, and glandular development in response to treatment. Additionally, they noted improved fertility outcomes in their model with adMSC treatment. A key feature of their study was that their model revealed that the GFP labeled cells were able to differentiate into epithelial cells [98]. Another important animal study compared local or intravenous administration of either rat bmMSC or adMSC in a rat mechanical endometrial injury model [99]. Their findings suggested both approaches were able to improve endometrial regeneration as noted by increased endometrial thickness and decreased collagen deposition. They ultimately concluded that local injection of adMSC produced the most favorable response based on the degree of reduction of collagen deposition and increased endometrial thickness.

There has only been one small pilot clinical study evaluating the use of autologous adMSC transplantation in five infertile women with severe AS [100]. Liposuction was performed to isolate adMSC from collected adipose tissue followed by transcervical intrauterine infusion of these adMSC. Resumption of menstruation occurred in two of the five patients, and oligomenorrhea improved in three of the women. There was a clinically significant increase in endometrial thickness from 3.0 to 6.9 mm in response to treatment. Following five embryo transfers, only one patient conceived but miscarried at nine weeks gestation. No adverse reactions were noted, suggesting adMSC therapy may be a safe option for the treatment of thin endometrium and AS; however, larger controlled studies are obviously warranted.

## 4. Mesenchymal Stem Cells for Management of Ovarian-Related Infertility

Premature Ovarian Insufficiency (POI) is defined by accelerated ovarian aging based on diminished reserve and steroid hormone synthesis prior to 40 years old [101]. The underlying etiologies of POI are vast and include genetic predisposition, autoimmune, metabolic, iatrogenic, and infectious. Depletion of ovarian reserve, whether physiologic (diminished ovarian reserve with aging), pathologic (POI), or gonadotoxic (chemotherapy), represents a significant contributor to infertility with poor response to ovarian stimulation for IVF.

In addition to endometrial regeneration, stem cells have promising potential for fertility preservation and the treatment of ovarian-related infertility. The potential for mitigating POI or chemotherapy-induced ovarian depletion has been explored with several animal models with proof of concept for either intravenous or direct intra-ovarian injection of human eMSC [102], MenSC [103,104], bmMSC [105,106,107], UC-MSC [108,109,110,111], amniotic MSC [112,113,114], and adMSC [115,116,117]. In most of the animal models, POI is induced with chemotherapeutic agents, such as cyclophosphamide and busulfan, although some models have used naturally aging ovarian models or surgical termination of ovarian function. Proposed mechanisms in which stem cells may restore ovarian function are enhanced folliculogenesis and neovascularization, decreased granulosa cell apoptosis, and restoration of ovarian steroid hormone production [118]. In a key translational study, Herraiz et al. found that human-derived bmMSC injected into immunodeficient mice with xenografted human ovarian tissue from poor responders improved follicular growth and vascularization [105]. While the exact mechanisms for restored folliculogenesis with stem cell administration are not known, some have theorized that it is in part secondary to reduced inflammatory injury and repletion of granulosa cell populations in response to stem cells, which support follicular development and steroid hormone production [119,120].

A large majority of clinical studies thus far were performed evaluating the use of bmMSC in the setting of POI. Gabr et al. presented an abstract with one of the earliest reports of laparoscopic ovarian autologous bmMSC injection in 30 women in Egypt less than 40 years of age with POI [121]. Although not published in a peer-reviewed journal, their abstract noted improved markers of ovarian reserve four weeks after treatment with reduced FSH, increased estrogen, and anti-mullerian hormone (AMH). One patient spontaneously conceived and three pursued IVF, but pregnancy outcomes were not reported [121]. Edessy et al. performed laparoscopic ovarian injection of autologous bmMSC in ten women mean age 26–33 years with idiopathic POI, and two out of ten patients resumed menses following treatment [122]. One patient spontaneously conceived and reported the first live birth following this treatment. Herraiz et al. evaluated a role for autologous bmMSC treatment via ovarian intra-arterial catheterization in 15 poor ovarian responders ages 34–37 with POI [123]. They reported improved antral follicle counts (AFC) and AMH levels two weeks following treatment. In their study, they noted improved outcomes with controlled ovarian stimulation for IVF although with a low euploidy rate (16%). Five patients conceived, two via IVF and three spontaneously [123]. *Gupta* et al. reported laparoscopic ovarian instillation of autologous bmMSC in a 45-year-old peri-menopausal woman, not in the setting of POI [124]. The patient’s AMH rose from 0.4 to 0.9 ng/mL eight weeks following the procedure. Three oocytes were retrieved with the development of one day-three embryo, which was frozen with a subsequent embryo transfer cycle that resulted in a live birth.

In addition to bmMSC, other stem cell populations have been studied in small clinical trials. In a study of poor ovarian responders, patients pursuing IVF received either autologous transvaginal ovarian injection of MenSC (n = 15) or standard treatment (control group, n = 16) [125]. The average age of patients was 35 and 36 years in the treatment and control groups, respectively. MenSC were isolated from plastic-adherent colony-forming cells plated on cycle day two of menses with confirmation of the presence of mesenchymal markers CD73, CD44, and CD90 and absence of hematopoietic marker CD45. They injected 150 microliters of dilute human serum albumin containing 20 × 10^6^ cells/mL directly into the left ovary using a transvaginal approach. Interestingly, there was no significant change in AMH, AFC, or oocytes retrieved in response to treatment, but fertilization and embryo development rates were higher with MenSC treatment. Over 50% (7/15) of the MenSC patients conceived with five live births, while 2/16 conceived in the control group with one live birth [125].

Mashayekhi et al. investigated safety outcomes of autologous ovarian adMSC transplantation via transvaginal ultrasound-guided injection in nine women with POI with three different cell count suspensions [126]. Women with an average age of 32 years old underwent liposuction of the sub-abdominal fat pads, yielding 50 mL of adipose tissue. The latter was washed, digested with collagenase, and purified cells were confirmed to be of mesenchymal origin using flow cytometry for analysis of CD105, CD90, and CD73. They found no significant associated treatment complications as their primary outcome. Their secondary outcomes included resumption of menses and ovarian reserve marker levels. Four of the nine women reported resumption of menses. No significant differences were reported with respect to ovarian reserve markers or ovarian volume parameters [126].

In addition to reduced ovarian reserve, oocyte quality significantly diminishes with age with increased risk for embryonic aneuploidy [127]. While stem cells may reduce inflammation, improve granulosa cell function and follicular development, their impact on age-related chromosomal errors contributing to aneuploidy are unknown. Stem cells may modulate the environment they inhabit or regenerate the tissue parenchyma, but they would not be expected to improve chromosomal errors contributing to the age-related risk for fetal aneuploidy. These clinical studies, while small, have begun to show an increased interest in potential utility and safety for the use of stem cells in the setting of diminished ovarian reserve or POI. However, larger, high-quality controlled trials are necessary before implementing these experimental treatments on a larger scale.

## 5. Challenges and Future Directions

While known causes for endometrial dysfunction, such as AS and atrophic/thin endometrium, have been outlined in this review, there remain many questions regarding how the endometrium contributes to infertility. Technologies such as preimplantation genetic testing of embryos have significantly decreased the potential for embryonic aneuploidy at embryo transfer, yet only up to 50% of euploid embryos implant, highlighting a critical role for better understanding the endometrium and initiating targeted therapies [128]. Clinical implications for the use of endometrial and non-endometrial-derived MSC for infertility have been extensively outlined in this review (Figure 1). Having already occupied a niche in the endometrium, eMSC and MenSC are strong candidates for cell therapy in the setting of endometrial-driven infertility, such as AS or atrophy. However, based on animal and human studies and a longstanding history for their use in other disease models, non-endometrial-derived MSC also represent a viable alternative treatment option. Through their inherent capacity to influence the endometrial niche, MSC can promote tissue regeneration and modulation of inflammation, thus reducing scarring and fibrosis. These attributes make their use for the treatment of AS and persistent thin endometrium appealing and the clinical studies highlighted in this review speak to their promising potential; however, larger clinical trials are warranted.

As mentioned previously, there are no available clinical studies evaluating eMSC in the setting of infertility, possibly due to the higher accessibility and scalability of other stem cell sources and precedence for their use in other systems. Obtaining eMSC and MenSC for therapeutic use is both clinically feasible and practical. Endometrial biopsies are low-risk procedures performed without anesthesia in the office setting with minimal discomfort and potentially less risk than a bone marrow aspiration for bmMSC. Menstrual cups are a safe alternative to disposable tampons and pads [129] and allow for easy collection for isolation and purification of stem cells. Furthermore, placental and umbilical cord-derived stem cells utilize specimens that would otherwise be discarded.

While there are a vast array of MSC sources, standardization and safety in isolation practices for clinical use will be of the utmost importance. Current good manufacturing practice (cGMP) must be applied to the design of clinical use of stem cell therapeutics for infertility. The intent of cGMP is to ensure the safety and reproducibility of treatments. Among these guidelines, the use of xeno-free media remains a significant challenge as MSC were previously cultured in fetal calf serum with associated heterogeneity in its contents. Further, the use of human-derived components (such as serum) displays similar heterogeneity and may induce differentiation towards a stromal fibroblast phenotype reducing their efficacy [56]. Infertility is a unique clinical field where the laboratory plays a central role, and moving research from the bench to the bedside is familiar territory. Once safety and reproducibility of MSC for the treatment of infertility is ensured, implementation could be easily streamlined through the clinical embryology laboratory.

Within the field of infertility, there are other future, although theoretical, potential outlets for the therapeutic use of MSC beyond endometrial regeneration. Clinical conditions such as endometriosis and adenomyosis have been associated with altered phenotypic characteristics of both eMSC [130] and MenSC [131]. Similar to performing bone marrow transplantation, functionally inadequate, but normal-appearing endometrium could theoretically be replaced by transplantation derived from either donor endometrial MSC or genetically modified autologous transplantation. While this indeed sounds like a far stretch now, it may not be so in reality, as we witness first-hand the rapid progress of technology, bioengineering, and gene editing.

Growing evidence suggests defective endometrial decidualization and subsequently, embryo implantation may contribute to the development of pregnancy complications such as abnormal placentation, placenta previa, or accreta [132]. These conditions are associated with significant maternal morbidity and their incidence increases with cesarean deliveries, perhaps due to abnormal endometrial regeneration at the hysterotomy site. Additionally, defective implantation and placentation have been postulated to contribute to the development of pre-eclampsia [133]. Albeit secondary, but nevertheless important, stem cell therapy could improve the “quality” of implantation through enhancing decidualization and potentially decrease the occurrence of conditions associated with abnormal placentation. Both infertility and high-risk pregnancy clinicians share the goal of reducing infertility and associated pregnancy-related risks due to endometrial dysfunction with the prospect of improving outcomes for hopeful mothers and their future babies.

## Figures and Tables

**Figure 1 jpm-11-01364-f001:**
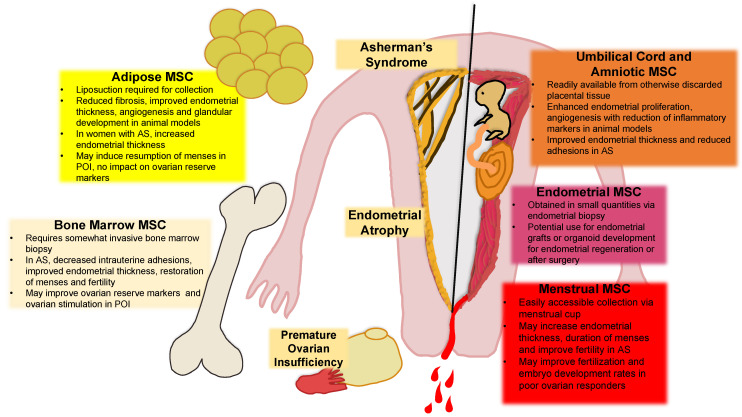
Sources of mesenchymal stem cells and implications for treatment of endometrial and ovarian-related infertility. This figure describes the various potential sources of mesenchymal stem cells including their collection and described therapeutic effects. The solid black line delineates physiologic endometrium (right) from pathologic endometrial environment (left). AS = Asherman’s Syndrome; MSC = Mesenchymal Stem Cell; POI = Premature Ovarian Insufficiency.

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
