# Peer review of "Making More Womb: Clinical Perspectives Supporting the Development and Utilization of Mesenchymal Stem Cell Therapy for Endometrial Regeneration and Infertility"

_jpm, 2021, doi:10.3390/jpm11121364_

Round 1
Reviewer 1 Report
In introduction: this manuscript has relied on efficiency of stem cell therapy on endometrial regeneration. Therefore, it should more paid on types of stem cells and the pros and cons of each stem cell in this section. Moreover, the mechanisms have been accepted about MSCs effect including anti-inflammatory, anti-apoptotic, anti-fibrotic, angiogenic effects, etc. should be regarded in introduction to rationalize their effectiveness in AS syndrome.
In section: 2.2. Persistent Thin Endometrium and Endometrial Atrophy
High dose estrogen is one of current strategies given to promote endometrial regeneration in Asherman syndrome, while it is neglected in this section.
Section 2.2, Line 112: Authors have mentioned Autologous platelet rich plasma treatment as a promising therapeutic for thin endometrium. In the actuality, based on some paracrine effects been attributed to PRP as well as stem cell therapy, it needs more discussion especially in comparison with characteristics and therapeutic potential of stem cell therapy in AS syndrome.
Section 2.2, Line 114: [unpublished data] is not required, while another available reference is cited.
In section: 3. Clinical Foundation for Stem Cell Use for the Treatment of Endometrial-related 120 Infertility
Line 124: The authors should pay to PRP as supportive treatment more prudently, while it is not a definite and approved treatment.
Page 6, line 230: The main probable mechanisms involved in efficiency of MSCs therapy in AS syndrome are their anti-inflammatory, anti-apoptotic and anti-fibrotic effects that are not related to OCT-4 expression. Meanwhile, OCT-4 is absent in MSCs like BM-MSCs.
The POR population was assessed about efficiency of stem cell therapy by Herraiz et al (ref.100) was 15 women as same as the POR population was evaluated by Zafardoust et al (ref 102) and last authors made effort to enroll close group. So, making judgement as “it is difficult to attribute the differences in outcomes to the treatment alone” only about last paper is not rational and should be corrected.
It is recommended that the authors in the beginning of each section explaining about potential of various stem cells firstly address the characteristics of each stem cell including immnunophentyping, proliferation ability,….
Moreover, the mechanisms have been accepted about MSCs effect including anti-inflammatory, anti-apoptotic, anti-fibrotic, angiogenic effects, etc. should be regarded in introduction to rationalize their effectiveness in AS syndrome.
In challenges: the authors should mention to limitation of MSCs culture in clinical setting under GMP standardization.
Author Response
The authors appreciate the reviewers’ constructive comments about our manuscript. Please find below our responses, following each specific comment.
- In introduction: this manuscript has relied on efficiency of stem cell therapy on endometrial regeneration. Therefore, it should more paid on types of stem cells and the pros and cons of each stem cell in this section. Moreover, the mechanisms have been accepted about MSCs effect including anti-inflammatory, anti-apoptotic, anti-fibrotic, angiogenic effects, etc. should be regarded in introduction to rationalize their effectiveness in AS syndrome.
Response: In developing this review, we hoped to provide clinical context with an emphasis on the translational and clinical studies thus far evaluating a role for stem cell use in the setting of infertility. In this Special Issue, there have been other reviews comparing the different types of stem cells, and we have done so throughout the body of the manuscript from the clinical perspective. In regards to the reviewer’s comment, we feel that comparing the different stem cells within the Introduction section would de-emphasize the main purpose of the review, since the purpose of the review is to highlight the need for solutions to these challenging clinical dilemmas,. Nevertheless, the identification of MSC and mechanisms for stem cell action for a therapeutic benefit have been highlighted within the Introduction section as suggested.
- In section: 2.2. Persistent Thin Endometrium and Endometrial Atrophy - High dose estrogen is one of current strategies given to promote endometrial regeneration in Asherman syndrome, while it is neglected in this section.
Response: We have added additional information regarding the use of estradiol in this section.
- Section 2.2, Line 112: Authors have mentioned Autologous platelet rich plasma treatment as a promising therapeutic for thin endometrium. In the actuality, based on some paracrine effects been attributed to PRP as well as stem cell therapy, it needs more discussion especially in comparison with characteristics and therapeutic potential of stem cell therapy in AS syndrome.
Response: We appreciate the reviewer’s comment. While the authors agree that the paracrine effects of PRP are interesting, comparing treatment effects of PRP (and other experimental therapies) versus stem cells is out of the scope of the current review and may confuse the reader.
- Section 2.2, Line 114: [unpublished data] is not required, while another available reference is cited.
Response: We have removed this as suggested by the reviewer.
- In section: 3. Clinical Foundation for Stem Cell Use for the Treatment of Endometrial-related 120 Infertility - Line 124: The authors should pay to PRP as supportive treatment more prudently, while it is not a definite and approved treatment.
Response: We have modified this section to highlight PRP along with the other mentioned therapeutics as proposed treatment but not standard of care.
- Page 6, line 230: The main probable mechanisms involved in efficiency of MSCs therapy in AS syndrome are their anti-inflammatory, anti-apoptotic and anti-fibrotic effects that are not related to OCT-4 expression. Meanwhile, OCT-4 is absent in MSCs like BM-MSCs.
Response: While OCT-4 is absent in most MSC (BM, adipose and placental derived), studies have shown OCT-4 positivity in MenSC to varying degrees but cytoplasmic as highlighted in Garget et al 2016. For clarity, we have included a statement describing these differences within the body of the manuscript.
- The POR population was assessed about efficiency of stem cell therapy by Herraiz et al (ref.100) was 15 women as same as the POR population was evaluated by Zafardoust et al (ref 102) and last authors made effort to enroll close group. So, making judgement as “it is difficult to attribute the differences in outcomes to the treatment alone” only about last paper is not rational and should be corrected.
Response: We have removed this statement as suggested.
- It is recommended that the authors in the beginning of each section explaining about potential of various stem cells firstly address the characteristics of each stem cell including immunophenotyping, proliferation ability,….
Response: As suggested, each section has been modified to contain this information.
- Moreover, the mechanisms have been accepted about MSCs effect including anti-inflammatory, anti-apoptotic, anti-fibrotic, angiogenic effects, etc. should be regarded in introduction to rationalize their effectiveness in AS syndrome.
Response: See response for #1.
- In challenges: the authors should mention to limitation of MSCs culture in clinical setting under GMP standardization.
Response: This has been added to the “Challenges and Future Directions” section as recommended.
Reviewer 2 Report
The paper is very actual in its expressed interest of improving endometrial function and overall fertility by use of stem cells. Its merit resides in investigating most of the relevant work in the field. But it fails to induce a directing line and to coagulate more focus in conclusion. Also the title is very promising and over ambitious, maybe a simpler 'stem cells use in restoring fertility' may be more appropriate.
Author Response
The authors appreciate the reviewers’ constructive comments about our manuscript. Please find below our responses, following each specific comment.
- The paper is very actual in its expressed interest of improving endometrial function and overall fertility by use of stem cells. Its merit resides in investigating most of the relevant work in the field. But it fails to induce a directing line and to coagulate more focus in conclusion. Also the title is very promising and over ambitious, maybe a simpler 'stem cells use in restoring fertility' may be more appropriate.
Response: We have significantly modified the conclusion section to focus on the feasibility, safety/reproducibility concerns for clinical use of MSC, future directions and implications for pregnancy. Thank you for the comments regarding the title. We have modified it to make it less ambitious.
Round 2
Reviewer 1 Report
All concern have been removed. Thus, the manuscript is accepted in current format.